

# Simulated Disperser Analysis: determining the number of loci required to genetically identify dispersers

Adam P.A. Cardilini[1], Craig D.H. Sherman[2], William B. Sherwin[3] and Lee A. Rollins[2,3]

[1] Faculty of Science, Engineering and Built Envrionment, Deakin University, Waurn Ponds, Vic, Australia
[2] Centre for Integrative Ecology, Deakin University, Waurn Ponds, Vic, Australia
[3] Evolution & Ecology Research Centre, School of Biological, Earth and Environmental Sciences, University of New South Wales, Sydney, NSW, Australia

## ABSTRACT

Empirical genetic datasets used for estimating contemporary dispersal in wild populations and to correctly identify dispersers are rarely tested to determine if they are capable of providing accurate results. Here we test whether a genetic dataset provides sufficient information to accurately identify first-generation dispersers. Using microsatellite data from three wild populations of common starlings (*Sturnus vulgaris*), we artificially simulated dispersal of a subset of individuals; we term this 'Simulated Disperser Analysis'. We then ran analyses for diminishing numbers of loci, to assess at which point simulated dispersers could no longer be correctly identified. Not surprisingly, the correct identification of dispersers varied significantly depending on the individual chosen to 'disperse', the number of loci used, whether loci had high or low Polymorphic Information Content and the location to which the dispersers were moved. A review of the literature revealed that studies that have implemented first-generation migrant detection to date have used on average 10 microsatellite loci. Our results suggest at least 27 loci are required to accurately identify dispersers in the study system evaluated here. We suggest that future studies use the approach we describe to determine the appropriate number of markers needed to accurately identify dispersers in their study system; the unique nature of natural systems means that the number of markers required for each study system will vary. Future studies can use Simulated Disperser Analysis on pilot data to test marker panels for robustness to contemporary dispersal identification, providing a powerful tool in the efficient and accurate design of studies using genetic data to estimate dispersal.

Corresponding author
Adam P.A. Cardilini,
a.cardilini@gmail.com

# INTRODUCTION

Dispersal is a major contributor to population processes, and can strongly influence genetic diversity, local adaptation, speciation, levels of inbreeding, sexual selection and many other biologically important processes (*Clobert et al., 2012*). Knowledge of dispersal patterns can improve evaluation of the maintenance of source–sink population dynamics (*Walsh et al., 2012*), the identification of habitat corridors or barriers to movement

(*Dzialak et al., 2005*; *Kozakiewicz et al., 2009*), the determination of biological limitations to dispersal (e.g., sex biased dispersal, *Goudet, Perrin & Waser, 2002*), and our understanding of how the environment affects dispersal behaviour (*Pruett-Jones & Lewis, 1990*). Such knowledge can contribute to a theoretical understanding of ecological processes and be applied to improve conservation management practices (*Fahrig & Merriam, 1994*; *Selonen, Hanski & Painter, 2010*; *Keller et al., 2010*). For over a century, direct measures of dispersal have been collected using mark-recapture data (*Sandercock, 2003*), and more recently using telemetry techniques (*Harris et al., 1990*) and Geographic Position Systems (GPS) tracking (*Cagnacci et al., 2010*). Unfortunately, these methods have practical constraints, including expense and the difficulty of deploying and retrieving a large number of units. Alternatively, genetic methods that identify individual contemporary dispersers are commonly used because they are relatively cheap and easy to implement (*Broquet & Petit, 2009*). These methods include Bayesian statistics to identify the probability that a particular genotype arises from each of a group of populations (*Pritchard, Stephens & Donnelly, 2000*; *Piry et al., 2004*; *Jombart et al., 2008*). If an individual's genotype has alleles that are more common to a population other than where it was sampled, the individual is considered to be a potential disperser (*Rannala & Mountain, 1997*).

The accuracy of genetic assignment methods depends on the demographic parameters of the species being investigated (e.g., rates of dispersal; *Bohonak, 1999*; *Whitlock & McCauley, 1999*), experimental design (e.g., proportion of populations sampled, number of individuals sampled; *Cornuet et al., 1999*; *Paetkau et al., 2004*; *Schwartz & McKelvey, 2008*), and the markers chosen (e.g., levels of polymorphism, number of loci; *Berry, Tocher & Sarre, 2004*). Given the number of factors that can influence the correct identification of dispersers using genetic assignment techniques, it is important that studies robustly test if the marker system and population structure allow for reliable identification and assignment of dispersers to the populations being studied. For example, *Berry, Tocher & Sarre (2004)* were able to determine the number of loci needed to accurately assign dispersers of grand skink (*Oligosoma grande)* populations by directly comparing genetically identified dispersers with mark-recapture data collected over a seven year period. Ideally, estimates of dispersal between populations should be carried out using multiple approaches to confirm congruence between estimation methods; however, most ecological studies are restricted to using a single approach either due to logistical or budget constraints (or both). Genetic data may permit the identification of dispersers and assignment to their source populations, but also provide the ability to estimate a range of other population parameters often of interest to ecological researchers (e.g., estimates of genetic diversity and heterozygosity, levels of inbreeding, multiple paternity, and historical patterns of connectivity; *Excoffier, Laval & Schneider, 2005*; *Peakall & Smouse, 2006*; *Porras-Hurtado et al., 2013*; *Waser & Hadfield, 2011*; *Waser, Paetkau & Strobeck, 2001*).

Here we perform a Simulated Disperser Analysis, using microsatellite data from three Australian common starling (*Sturnus vulgaris*) populations to test the power of a marker system to accurately identify 'dispersers' (individuals that are not native to their collection locality) using genetic assignment tests and genetic data only. This approach simulates dispersal between populations and then tests a number of parameters (number of loci,

levels of variability of loci, and effect of disperser genotype) on the ability to identify dispersers and to correctly assign these to their collection locality.

## METHODS

### Ethics statement

The collection of all samples followed the strict guidelines outlined in the ethics application (permit number: 05/011A) approved by The University of New South Wales ethics committee.

### Sample collection and DNA methods

We investigated dispersal in Australian invasive starlings (*Sturnus vulgaris*), a highly vagile species that maintains an extensive distribution across southeast Australia. Samples were taken from three populations previously shown to have low levels of genetic differentiation (*Rollins et al., 2009*; $F_{ST} = 0.02–0.07$): Orange (New South Wales, 33°17′S, 149°06′E, $N = 32$); Mallala (South Australia, 34°27′S, 138°30′E, $N = 32$); and Munglinup (Western Australia, 33°42′29″S, 120°51′54″E, $N = 30$).

DNA was extracted using Gentra PureGene DNA extraction kit (Qiagen) following the manufacturer's instructions. Microsatellites were developed using next generation sequencing on the GS-FLX 454 platform (Roche, Manheim, Germany) following methods described by *Abdelkrim et al. (2009)*. QDD v 0.9.0.0 Beta (*Meglécz et al., 2010*) was used to identify microsatellites from the sequencing data and primers were designed using the program PRIMER 3 (*Rozen & Skaletsky, 2000*). A panel of 20 polymorphic markers was chosen (see Table S1). PCR reactions were multiplexed using universal fluorescently labelled primers (*Neilan, Wilton & Jacobs, 1997*). A step-down PCR protocol was used for each reaction consisting of ten cycles each at the following annealing temperatures: 70 °C, 64 °C, 58 °C, 54 °C, 50 °C. Reactions containing DNA from the same individual but with differently labelled universal markers (e.g., PET, NED, 6-FAM and VIC) were combined in equivalent amounts, so that all loci for each individual could be multi-loaded into three tubes prior to fragment analysis (see Table S1). Samples were genotyped using an ABI 3730 (Applied Biosystems, Foster City, CA, USA) using GS-500 (Liz) in each capillary as a size standard. Allele sizes were estimated on GENEMAPPER version 3.7 (Applied Biosystems). These data were combined with data from 11 microsatellite loci for the same individuals as detailed in *Rollins et al. (2009)* giving us a total of 31 loci.

Microsatellite markers were checked for departures from Hardy-Weinberg equilibrium (Arlequin version 3.5.1.2, *Excoffier, Laval & Schneider, 2005*) and linkage disequilibrium (GenePop version 4.0.10, *Rousset, 2008*). PIC was calculated for each locus using PICcalc (*Nagy et al., 2012*). Arlequin was used to calculate Pairwise $F_{ST}$ values. Pairwise values for Shannon's mutual information index (I, formerly called $^{S}H_{UA}$, *Sherwin et al., 2017*) were calculated in GenAlEx (*Peakall & Smouse, 2006*) because mutual information is better than $F_{ST}$ at handling a range of population sizes and dispersal rates (*Sherwin et al., 2006*; *Dewar et al., 2011*).

Ninety-four samples were successfully genotyped for 31 microsatellite loci. Of the 20 new species-specific loci developed here, two showed significant deviation from Hardy-Weinberg equilibrium (*Svu002* and *Svu010*) and were removed from further analyses (see Table S3). None of the remaining 18 loci showed significant departures from linkage equilibrium after Bonferroni correction. Data from the other 11 loci previously have been shown to be in Hardy-Weinberg equilibrium in these populations by *Rollins et al. (2009)*.

## Simulations

Initially, we used GeneClass2 (*Piry et al., 2004*) to determine the probability of each individual originating in the population where it was sampled (see method below). One individual from each population was randomly assigned to be a 'simulated disperser'; individuals with low probabilities of population membership were excluded from being simulated dispersers because they were potentially real dispersers. Simulated dispersers were moved in every possible combination relative to other simulated dispersers creating a total of 27 ($3^3$) treatments (including a treatment of no movement, Table 1).

Loci were ranked by their Polymorphic Information Content (PIC) (see *statistical analysis below*). Each treatment was analysed using 57 genetic datasets. The first analysis started with the locus with the highest PIC and then consecutively adding the next highest PIC locus until all loci were added ($N = 29$, including the dataset having all markers; referred to as highest PIC hereafter). The second analysis started with the locus with the lowest PIC and then consecutively adding the next lowest PIC locus until all loci were added ($N = 28$, because the dataset with all markers was analysed in the previous analysis; referred to as lowest PIC hereafter).

The partial Bayesian method (*Rannala & Mountain, 1997*) implemented in GeneClass2.0 was used to detect simulated dispersers. We ran 10,000 MCMC simulations per population using the $L_h/L_{max}$ likelihood computation (*Paetkau et al., 2004*). Individuals were considered first-generation dispersers if they had an $L_h/L_{max}$ p-value that was below 0.01. An individual can incorrectly be identified as a disperser in two ways: if they are identified as a disperser when they are not one (False Positive), or, when they fail to be identified as a disperser when they are one (False Negative). Once identified, GeneClass2.0 assigns dispersers to the most likely source population.

In our assessment of the ability of GeneClass2.0 to detect simulated migrants, assignment tests were repeated for treatments and individuals, resulting in the non-independence of data. To account for this, we used Generalised Linear Mixed Models (GLMMs), specifying treatment and individual as random effects, and predictor variables (number of loci, highest/lowest PIC loci used, location the simulated disperser was moved to, and I) as fixed effects (*Zuur et al., 2009*). The response variable 'dispersal status correctly identified' was a binary indicator of the ability of GeneClass2.0 to correctly identify whether a simulated disperser was a disperser or not (0 = incorrectly identified, 1 = correctly identified) and was specified as having a binomial distribution of errors with a logit link function. For example, if an individual was identified as a resident in the population from which it was sampled, it was classed as 'dispersal status correctly identified'. The response variable (dispersal status correctly identified) was modelled as a function of (a) the number of

**Table 1 List of simulated disperser movement combination used in analysis.** Simulated dispersers were moved in every possible combination relative to other simulated dispersers creating a total of 27 ($3^3$) treatments. This table indicates the location of each simulated disperser for each of the 27 treatments. Treatment 0 is the collection locality of each of the simulated dispersers before any simulated movement has taken place. For example, for treatment 0, simulated disperser A is located in Munglinup, B is located in Mallala and C in Orange (their collection localities). In treatment 1 simulated disperser A stayed in Munglinup while B and C were both moved from their collection localities (treatment 0) to Munglinup.

| Treatment | Munglinup | Mallala | Orange |
|---|---|---|---|
| 0 | A | B | C |
| 1 | ABC | 0 | 0 |
| 2 | 0 | ABC | 0 |
| 3 | 0 | 0 | ABC |
| 4 | AB | C | 0 |
| 5 | AB | 0 | C |
| 6 | AC | B | 0 |
| 7 | AC | 0 | B |
| 8 | BC | A | 0 |
| 9 | BC | 0 | A |
| 10 | 0 | BC | A |
| 11 | A | BC | 0 |
| 12 | A | 0 | BC |
| 13 | 0 | A | BC |
| 14 | 0 | AB | C |
| 15 | C | AB | 0 |
| 16 | 0 | AC | B |
| 17 | B | AC | 0 |
| 18 | 0 | C | AB |
| 19 | C | 0 | AB |
| 20 | B | 0 | AC |
| 21 | 0 | B | AC |
| 22 | C | A | B |
| 23 | B | C | A |
| 24 | C | B | A |
| 25 | A | C | B |
| 26 | B | A | C |

loci, (b) whether the highest/lowest PIC loci were used, (c) the population in which the simulated disperser was located (either its collection locality or one of the other two populations) and (d) the pairwise I value between populations when moved. The number of loci was a discrete numeric variable, while highest / lowest PIC loci used (PIC range = 0.251–0.862) was specified as a categorical variable with two levels, location was specified as a categorical variable where individuals were coded as either being from their collection locality or one of the other two populations (collection locality was used as the reference group with which other groups were compared), and pairwise I was specified as a continuous numeric variable. Model fit was estimated from marginal ($R^2_{\mathrm{GLMM/(m)}}$) and conditional ($R^2_{\mathrm{GLMM/(c)}}$) coefficients of determination, following *Nakagawa & Schielzeth*

*(2013)*. $R^2_{\text{GLMM}/(m)}$ estimates model fit using fixed effects only, while $R^2_{\text{GLMM}/(c)}$ estimates model fit including both fixed and random effects. By comparing these estimates it is possible to compare the contribution that random effects and fixed effects have on the response variable.

### Literature search

To assess how studies have addressed parameters that affect power to detect dispersers (number of individuals, number of loci, levels of genetic differentiation) we used Google Scholar to conduct a literature search (on 19/12/2016) for articles that cited the publication that originally presented the GeneClass2 analysis (*Piry et al., 2004*). We used the key term 'dispersal' to narrow the search and of 1,280 articles identified, we randomly chose 160 articles to assess. If the article did not use GeneClass2.0's migrant detection analysis we disregarded it; if the article did use this method we collected information on the number of loci used, global $F_{\text{ST}}$ value, and general article meta-data such as author(s), year published, common name and species name. Of the 160 papers that were assessed, 132 matched the selection criteria, including a total of 136 datasets (Table S2).

Generalised Linear Models (GLMs) were used to determine the relationship between the global $F_{\text{ST}}$ values of studies using genetic assignment tests and the number of loci that were used by these studies to identified dispersers. The number of loci used was specified as a continuous response variable, with a Poisson distribution, which was determined graphically and the global $F_{\text{ST}}$ value was a fixed effect.

To demonstrate the continuing relevancy of this work (see *De Barba et al., 2016*), we searched within the 1,280 articles we identified above for those containing the search term 'microsatellite' by year since this technique was published (2005–2016) to determine whether the number of papers using this approach was decreasing over time.

### Statistical analysis

Regression models and graphics were generated using the statistical programming environment R (version 2.15.2, *R Development Core Team, 2012*). GLMMs were generated using the function glmer within the package lme4 (version 0.999999-2, *Bates, Maechler & Bolker, 2012*). Graphics were generated using the package ggplot2 (version 0.9.3.1, *Wickham, 2009*).

## RESULTS

### Sample inclusion and genetic analysis

Populations from Orange (New South Wales) and Mallala (South Australia) showed the lowest levels of differentiation from each other ($F_{\text{ST}} = 0.026$, $P$ value $< 0.001$; $I = 0.081$), while the population from Munglinup (Western Australia) displayed higher levels of genetic differentiation from Orange ($F_{\text{ST}} = 0.082$, $P$ value $\leq 0.001$; $I = 0.152$) and Mallala ($F_{\text{ST}} = 0.081$, $P$ value $\leq 0.001$; $I = 0.139$) indicating that Munglinup is the most genetically distinct area sampled.

One of 94 individuals was identified as a potential disperser within the original dataset, which included all loci and no simulated dispersal. We avoided using this individual as a simulated disperser.

**Table 2  Results of model testing the impact of variables on correctly identifying simulated dispersers.** Genetic assignment tests were used to determine whether three common starlings (*Sturnus vulgaris*) from three genetically distinct collection localities would be identified as dispersers when their movement to a new location was simulated. Given are results from a Generalised Linear Mixed Model used to determine the effect of the predictor variables, the number of loci, loci with high or low Polymorphic Information Content (PIC) and simulated disperser movement, on the ability to correctly identify the status of simulated dispersers. The relationship of the response variable (simulated disperser's status correctly identified by a genetic assignment test) was tested by specifying movement in two ways. The first model specified movement as the location to which a simulated disperser was moved (i.e., 'In Mallala' means a simulated disperser was moved from its collection locality to Mallala). Genetic distance (I) between the two locations (collection locality and simulated dispersal location) was used as a predictor variable in the second model. C, coefficient; SE, standard error; Z, test statistic. Model fit was estimated from marginal ($R^2_{\mathrm{GLMM/(m)}}$) and conditional ($R^2_{\mathrm{GLMM/(c)}}$) coefficients of determination; $R^2_{\mathrm{GLMM/(m)}}$ estimate model fit using fixed effects only, while $R^2_{\mathrm{GLMM/(c)}}$ estimates model fit including both fixed and random effects.

| Response variable | Predictor variable | C | SE | Z |
|---|---|---|---|---|
| Simulated disperser status correctly identified by genetic assignment test | Number of loci | 0.30 | 0.01 | 26.98 |
| ($R^2_{\mathrm{GLMM/(m)}} = 0.6260$; $R^2_{\mathrm{GLMM/(c)}} = 0.8035$) | Highest/lowest PIC loci used | 1.32 | 0.11 | 11.62 |
| | Moved to Mallala | −4.75 | 0.24 | −19.91 |
| | Moved to Munglinup | −3.18 | 0.20 | −15.71 |
| | Moved to Orange | −5.75 | 0.25 | −23.31 |
| Simulated disperser status correctly identified by genetic assignment test | Number of loci | 0.38 | 0.01 | 25.39 |
| ($R^2_{\mathrm{GLMM/(m)}} = 0.7390$; $R^2_{\mathrm{GLMM/(c)}} = 0.8039$) | Highest/lowest PIC loci used | 2.18 | 0.14 | 15.07 |
| | I | 37.16 | 3.03 | 12.25 |

## Simulations

The simulation results showed that the ability to correctly identify a simulated disperser was strongly associated with an increase in the number of loci used and when the highest PIC loci were used rather than the lowest (Table 2, Fig. 1). For a subset of the simulation data (treatments where simulated dispersers were moved outside their collection locality), I values showed a strong positive relationship with correct identification of simulated dispersers. When a simulated disperser was moved to a location that had a relatively high pairwise I to its collection locality, it was more likely to be correctly identified as a disperser (Table 2). For two of our simulated dispersers, no False Positive errors were identified. However, False Negative errors were common.

### Simulated disperser A (SD-A), artificially moved from its collection locality at Munglinup

SD-A was always identified as a resident when located in its collection locality, whether using the highest or lowest PIC loci. We were able to identify SD-A to its collection locality when using the three highest PIC loci available, whereas 11 loci were needed when using those having the lowest PIC values. The minimum number of loci required to identify SD-A to its collection locality 100% of the time did not vary between locations (Fig. 2).

### Simulated disperser B (SD-B), moved from Mallala

SD-B was incorrectly identified as a disperser when located in its collection locality when using the highest PIC loci but was always identified as a resident when using the lowest PIC loci. When using the lowest PIC, 20 or more loci were needed to assign SD-B to its collection locality 100% of the time, while 27 loci were needed when using the highest

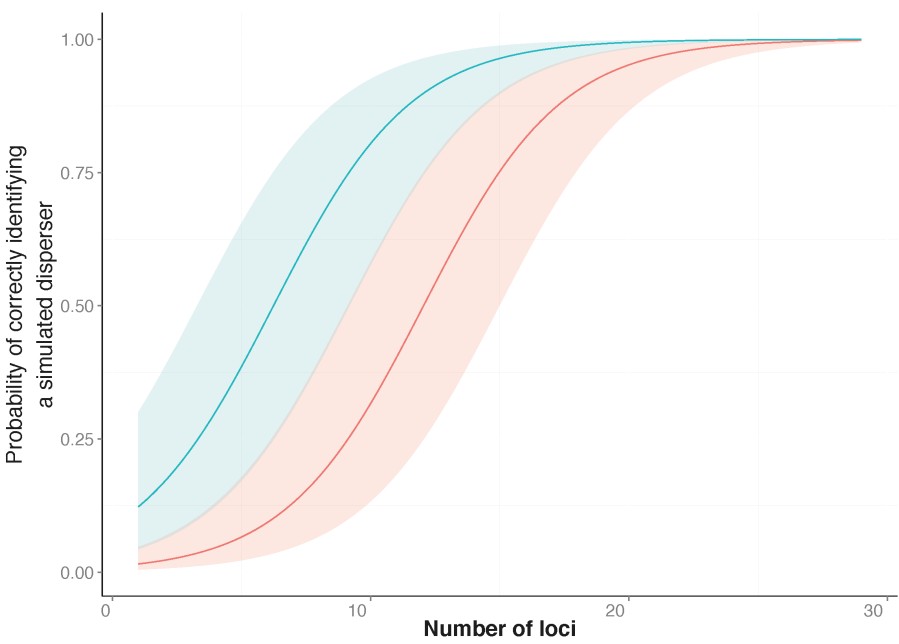

**Figure 1** **Probability of correctly identifying a simulated disperser by the number of loci used.** Genetic assignment tests were used to determine whether three common starlings (*Sturnus vulgaris*) from three genetically distinct collection localities would be identified as dispersers when their movement to a new location was simulated. Shown is the relationship between the probabilities (*y*-axis), from a Generalised Linear Mixed Model, of correctly identifying a simulated disperser plotted against an increase in the number of loci used. The top line indicates the response when using loci with the highest Polymorphic Information Content (PIC) first (blue line), and the bottom line indicates the response when using loci with the lowest PIC first (red line). The coloured shading corresponding to each line encompasses the upper and lower confidence intervals of the models.

PIC loci. SD-B was identified as a disperser after using the 18 highest PIC loci; however, detection broke down between 21 to 24 loci. When using the lowest PIC loci SD-B was not correctly assigned 1.66% of the time, and only when located in Orange. SD-B was not correctly assigned 14.94% of the time when using the highest PIC loci (Fig. 2).

### Simulated disperser C (SD-C), moved from Orange

SD-C was always identified as a resident when located in its collection locality whether using the highest or lowest PIC loci. Fifteen loci were required to identify SD-C to its collection locality 100% of the time, whether using the highest or lowest PIC loci. When using the highest PIC loci, 21 out of 27 treatments correctly identified SD-C with 10 or more loci. For treatments from 1 to 15 loci where SD-C was incorrectly identified, the location of SD-A contributed to the incorrect identification. SD-C was not correctly assigned to Mallala 2.81% of the time when using the highest PIC loci and was not correctly assigned only when located in Munglinup (Fig. 2).

### Literature search

The studies we evaluated were published between 2005 and 2016. Studies that implemented first-generation migrant detection in GeneClass2.0 to identify dispersers used on average
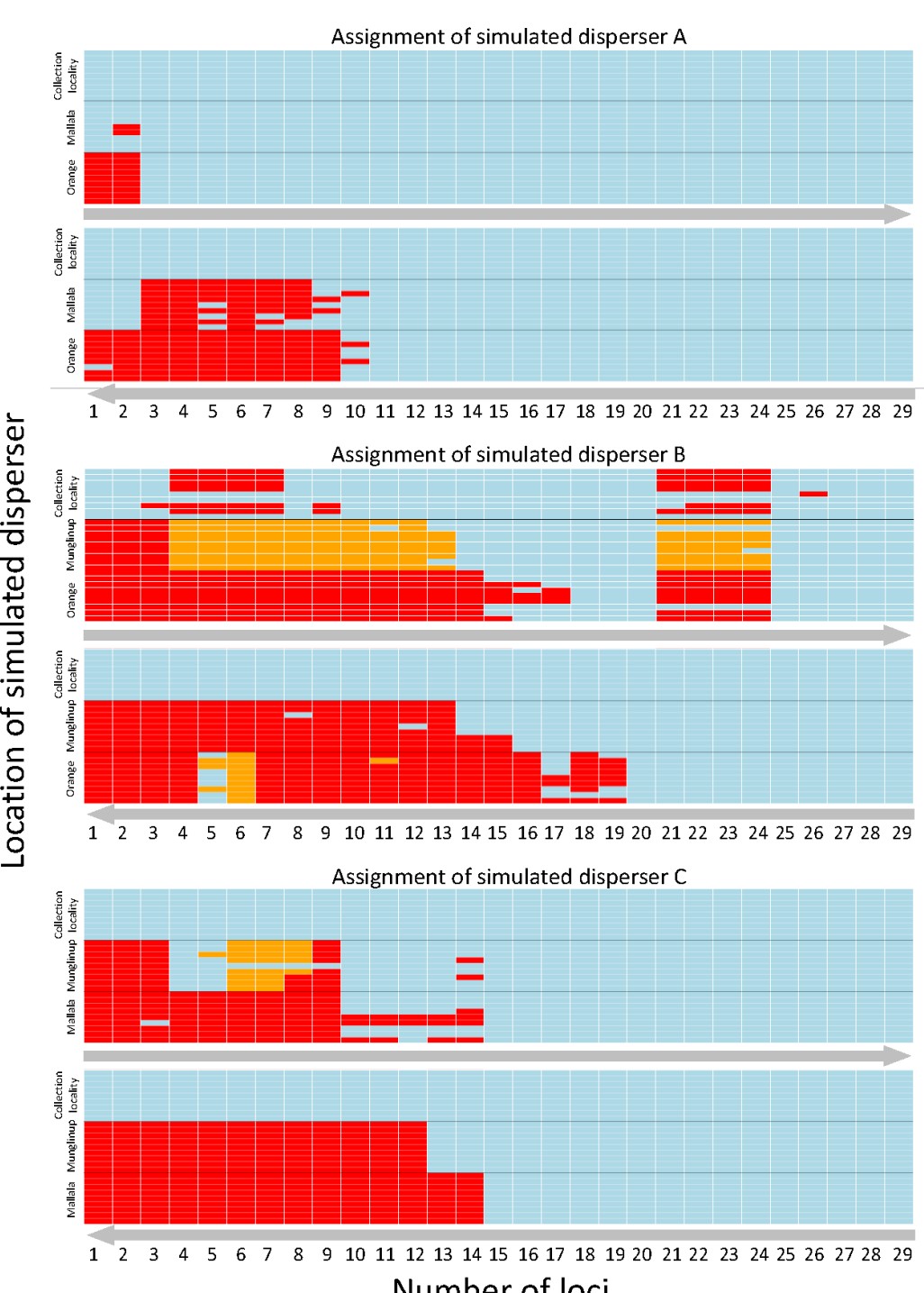

**Figure 2  Heatmap showing the assignment status of each simulated disperser.** Genetic assignment tests were used to determine whether three common starlings (*Sturnus vulgaris*) from three genetically distinct collection localities (A, Munglinup; B, Mallala; C, Orange) would be identified as dispersers when their movement to a new location was simulated. Each small coloured rectangle (continued on next page...)

**Figure 2 (…continued)**
represents the results of a genetic assignment test; blue, individual was correctly assigned to its collection locality; red, individual was not recognised as a disperser or resident; orange, individual was identified as a disperser but assigned to the incorrect collection locality. The three panels represent the results for simulated disperser A, C and B. Within each panel the top heat map shows the results when first using loci with the highest Polymorphic Information Content (PIC) and the bottom heat map shows the results when first using loci with the lowest PIC. The grey arrows below each inset show the direction the loci were sorted, from high PIC to low PIC (top) or vice versa (bottom). Each inset is sorted on the $y$-axis by the location the individual was found and the movement treatment. Each row corresponds to a different treatment. The treatment order of each simulated migrant's heat map row can be found in Table S4.

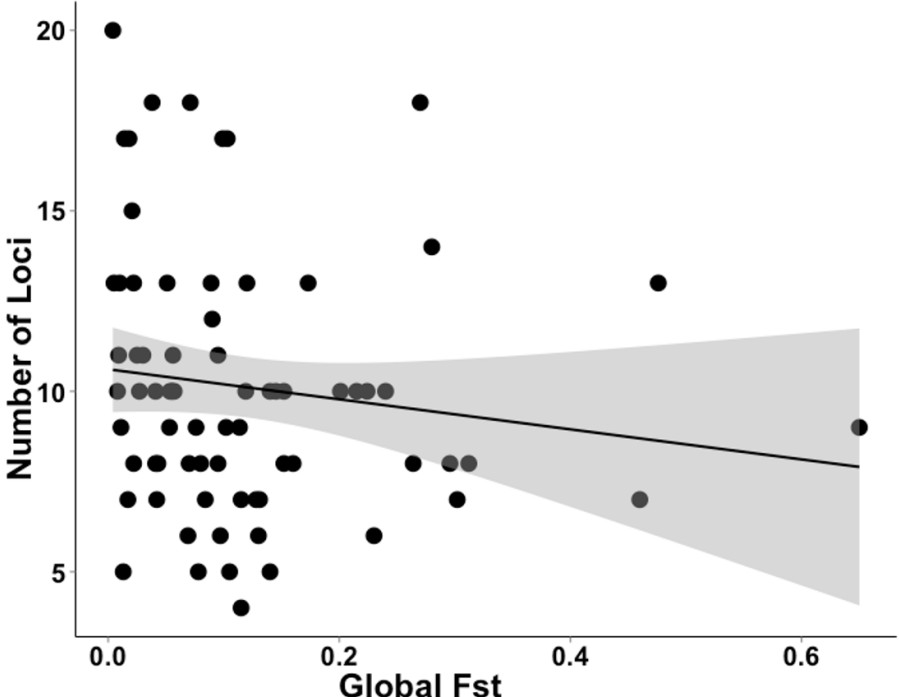

**Figure 3** **Data extracted during a literature review showing the relationship between the number of loci used and $F_{ST}$ value.** A literature review was conducted to gather data on studies that used genetic assignment tests in GeneClass2.0 and their global $F_{ST}$. This graph shows the non-significant relationship (GLM, $t_{71} = -1.312$) between the number of loci used and global $F_{ST}$ of 72 datasets. The grey band shows the 95% confidence interval on the fitted values.

11.2 ±5.1 loci. For a subset of these data ($N = 72$), where global $F_{ST}$ was available, the average global $F_{ST}$ was, 0.120 ±0.119. There was no relationship between the global $F_{ST}$ of these studies and the number of loci used when conducting genetic assignment tests (GLM, $t_{71} = -1.312$; Fig. 3).

Figure 4 demonstrates that, despite the recent increase in population genetic papers based on next-generation sequencing data, microsatellite data sets continue to be used for contemporary disperser identification at a similar rate over the past eight years (more than 100 papers per year during this period).

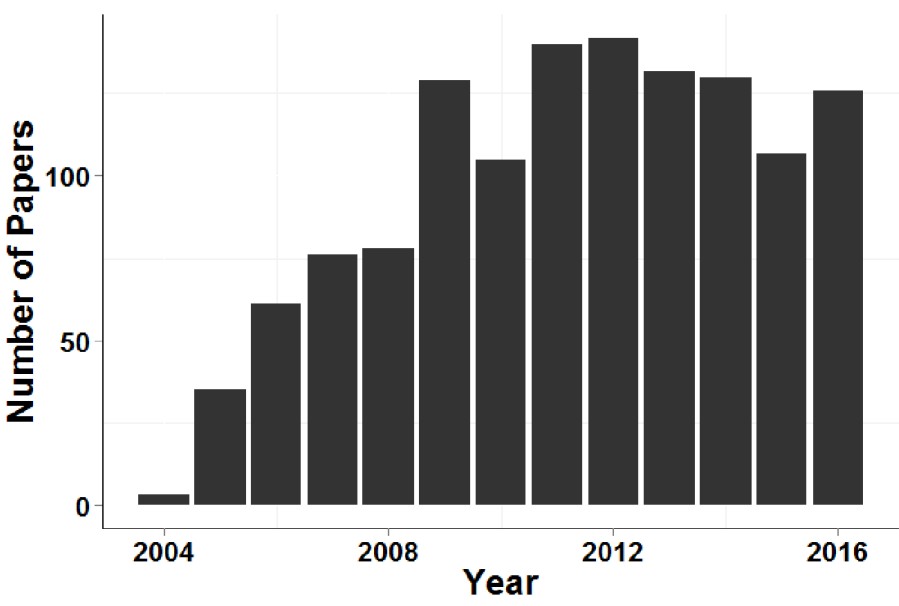

**Figure 4** Number of papers that cited *Piry et al. (2004)* and contained the search term 'microsatellite' each year between 2005 and 2016.

## DISCUSSION

Many studies have emphasised the need to have sufficient numbers of highly variable loci in order to carry out population assignment tests with a high degree of confidence (*Manel, Berthier & Luikart, 2002*; *Berry, Tocher & Sarre, 2004*; *Evanno, Regnaut & Goudet, 2005*). Here we artificially force dispersal between a set of populations and test the power of the marker system (i.e., number of loci and levels of variability) to identify dispersers. Our results show that we should be validating genetic assignment tests by testing the power of the marker systems we use. Without validation of genetic data sets we cannot be confident in the inferences we make about dispersal. Our results confirm that the number of loci is one of the best predictors that a marker system has adequate power to detect real dispersal events (*Smouse & Chevillon, 1998*; *Bernatchez & Duchesne, 2000*). However, both false positive (identified as a disperser when they are not one) and false negative (not identified as a disperser when they are one) assignments can result even when using many loci with high levels of polymorphism (e.g., simulated disperser B). The approach we present highlights the number of false positive and false negative assignments (see Fig. 2).

Similar to previous studies (*Bernatchez & Duchesne, 2000*; *Berry, Tocher & Sarre, 2004*), our data indicated that the number of loci used to identify dispersers strongly influences the results of contemporary genetic disperser analysis. As more loci are added to an analysis the total information available for discriminating between the genotype of an individual and the background population increases, which increases the detectability of dispersers. Others have suggested that approximately 10 loci are required to correctly identify dispersers when using assignment tests with microsatellite data (*Cornuet et al., 1999*; *Bernatchez & Duchesne, 2000*); however, for our study system, ideally 27 loci were required (Fig. 1).

In our study, false negatives (i.e., not identifying an individual as a migrant when they were) were more common than false positives (i.e., when an individual was identified as a migrant when they were not). This difference highlights the fact that every study requires an individual validation of the marker system in order to be confident of identifying dispersers using genetic assignment tests.

Not surprisingly, when using loci with low PIC more loci are usually required to reach the same level of discrimination than loci with high PIC because low PIC loci provide less discriminatory power for the analysis. However, this is not always true as seen with individual B when using 21 to 24 of the highest PIC loci. A single locus can result in inconsistent assignment of dispersers even when using a large number of loci. It is possible that a single locus can play a comparatively large role in distinguishing the multi-locus genotype of an individual. For example, if an individual has alleles at a locus that are rare in its population but more common in another population, inclusion of this locus may increase the incorrect assignment of that individual. Post-hoc analyses, where locus 21 was removed, resulted in correct assignment of SD-B for all locus combinations. This demonstrates how our approach can help to identify problematic loci (i.e., those that yield inaccurate assignments). Although spurious results might occur with any combination of loci, it is reasonable to assume that such results would generally become less likely as more discriminatory information becomes available with an increased number of loci.

Our results confirm those of previous studies (*Cornuet et al., 1999*; *Berry, Tocher & Sarre, 2004*) by showing that it was easier to identify a disperser when they were moved into a population that had a relatively high pairwise differentiation (e.g., I) from their collection locality. Although *Berry, Tocher & Sarre (2004)* clearly showed this effect in a natural system over a decade ago, there has been surprisingly little response to these findings. If researchers were to take into account the relationship between population differentiation (e.g., I, $F_{ST}$) and the number of loci required and the ability to correctly identify dispersers, studies that use genetic assignment tests should use more loci when their global $F_{ST}$ is lower. As shown by our analysis of the literature (Fig. 3), studies with lower $F_{ST}$ values are not consistently using more loci to compensate for detection ability. By not dealing with this relationship, published studies may have incorrectly identified dispersers, potentially leading to misleading conclusions. To be confident of research outcomes, we recommend that studies only using genetic assignment test to identify dispersers should employ an approach similar to the Simulated Disperser Analysis that we describe here to validate their marker panel.

For a quick and stringent method capable of determining the number of loci necessary to accurately identify dispersers (e.g., an approach that is able to distinguish individuals that are easy to identify and hard to identify, such as SD-A and SD-B respectively) a 'worst-case scenario' is necessary. This study showed that a worst-case scenario can be expected to occur when individuals are moved to the most genetically similar population and the lowest PIC loci are used. Therefore, for future studies to validate their disperser analysis, they could move simulated dispersers to the most genetically similar population, start with the lowest PIC locus and continue by adding the next lowest PIC locus (Fig. 5). Such a

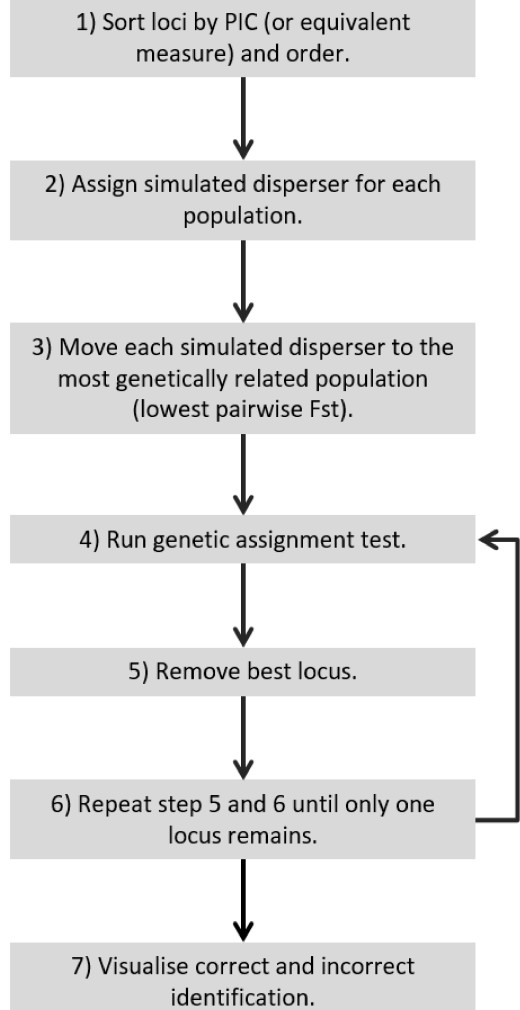

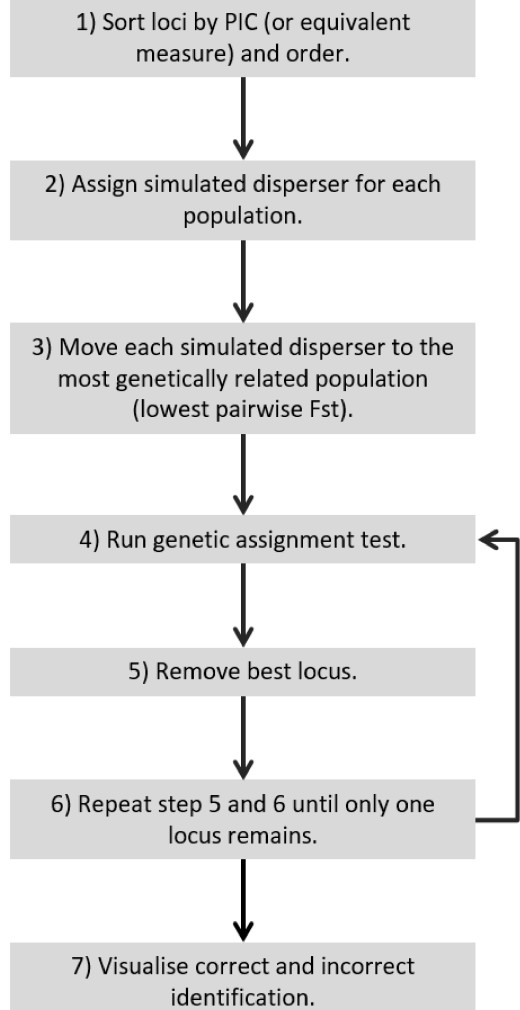

**Figure 5** Recommended workflow indicating the steps necessary for validating genetic assignment tests.

simulated dispersal scenario makes biological sense given that genetic populations that have the lowest pairwise $F_{ST}$ values are likely to be geographically closest to one another, and dispersers are more likely to move to a population closer to their own rather than further away (*Sutherland et al., 2000*). If spurious results are observed (false positive), such as when using the 21st locus to detect SD-B caused consistently incorrect results in our study, the offending loci may require greater scrutiny because this could result from mis-scoring.

In studies where some information on levels of genetic differentiation and levels of loci polymorphism are known, a preliminary "true" simulation study (i.e., based on synthetic samples) can potentially provide guidance on sample sizes needed to correctly identify dispersers. However, a simulated dispersal approach is applicable in cases where there is no prior information on levels of genetic differentiation and levels of loci polymorphism.

It is also important to note that a simulated dispersal approach will only be valuable when estimates of the underlying allele frequencies are gained from a robust sampling protocol.

## CONCLUSION

An increase in the number of loci and the pairwise differentiation (e.g., I, $F_{ST}$) between a disperser's collection locality and its final destination increases the accuracy of genetically assigning the disperser (*Cornuet et al., 1999*; *Bernatchez & Duchesne, 2000*; *Berry, Tocher & Sarre, 2004*). Our study showed similar patterns to previous research on natural systems (*Berry, Tocher & Sarre, 2004*), however, the magnitude of the effect varied. Although basic patterns may be discernible when determining the number of loci necessary to accurately identify dispersers, the unique nature of natural systems means that every study system will be different. By validating our genetic disperser analysis, we were able to determine the number of loci required to increase the accuracy of identifying dispersers in a wild population without the need for direct measures of dispersal (e.g., tracking data, etc). Although next-generation sequencing data sets are becoming more prevalent in population genetic studies, we demonstrate here that microsatellite data are still commonly used to estimate contemporary dispersal and believe they will continue to do so (*De Barba et al., 2016*). Although our demonstration of this approach used microsatellite data, the process is also applicable analyses of next-generation sequencing data sets.

## ACKNOWLEDGEMENTS

We thank the Ramaciotti Centre for Gene Function Analysis for 454 sequencing and also Simon Griffith for the use of his laboratory space. Thanks to Andrew Woolnough, Ron Sinclair, John Tracey and the WA Department of Agriculture and Food, for help collecting samples. We would like to acknowledge the starlings that were killed as part of a control program and used in this research.

### Funding
This work was funded by the Australian Research Council (#LP0455776) and a Deakin University fellowship to Lee A. Rollins. The funders had no role in study design, data collection and analysis, decision to publish, or preparation of the manuscript.

### Grant Disclosures
The following grant information was disclosed by the authors:
Australian Research Council: #LP0455776.
Deakin University.

### Competing Interests
Lee A. Rollins is an Academic Editor for PeerJ.

## Author Contributions

- Adam P.A. Cardilini conceived and designed the experiments, performed the experiments, analyzed the data, contributed reagents/materials/analysis tools, prepared figures and/or tables, authored or reviewed drafts of the paper, approved the final draft.
- Craig D.H. Sherman and William B. Sherwin conceived and designed the experiments, contributed reagents/materials/analysis tools, authored or reviewed drafts of the paper, approved the final draft.
- Lee A. Rollins conceived and designed the experiments, performed the experiments, analyzed the data, contributed reagents/materials/analysis tools, authored or reviewed drafts of the paper, approved the final draft.

## Data Availability

Cardilini, Adam PA, Sherman, Craig DH, Sherwin, William B, & Rollins, Lee A. (2018). Microsat Data for 'Simulated Disperser Analysis: determining the number of loci required to genetically identify dispersers' [Data set]. PeerJ. Zenodo. http://doi.org/10.5281/zenodo.1202093. The raw data is available in the Supplemental Files.

## Supplemental Information

Supplemental information for this article can be found online at http://dx.doi.org/10.7717/peerj.4573#supplemental-information.

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
