# Peer review of "Simulated Disperser Analysis: determining the number of loci required to genetically identify dispersers"

_PeerJ, doi:10.7717/peerj.4573_

## Round 0.1 · original submission · Minor Revisions

· Academic Editor

Minor Revisions

Overall both referees are encouraging about the approach and this contribution to the field. However both also point towards some missed opportunities to take this further and make it more quantitative. I can see their point, and it would make an interesting and worthwhile paper that much better if you were to take their suggestions and expand the approach beyond the "seat of the pants" framework to one that would apply more broadly as each of the referees suggest.

But, even if you decide not to expand it, both referees are still supportive of the manuscript, because as one states, even with the limitations, this approach would be better than the one taken now, which is basically to ignore the issue.

So, I leave it to you to decide. You have two outstanding referees on this submission, and I believe that they have offered you some excellent advice on how the approach could be expanded and improved for much broader utility and impact on the field (which ultimately translates to citations for you). However, if you decide not to expand this, and simply address the comments about clarity, literature search, and the limitations that your procedure cannot prevent or correct errors in the sampling protocol used to obtain the dataset outlined by the referees here, I expect that the manuscript will become acceptable to them for publication.

Reviewer 1 ·

Basic reporting

The manuscript is well written and easy to understand. However, there are some sentences that are not logical structured. For example:

lines 55-56: An individual genotype cannot be similar to a population. It can only be similar to another genotype randomly sampled from another population.

lines 157-159: I’m not sure I understand this. In line 154 it is stated that the explanatory variable is “the population in which the simulated disperser was located”. Although this statement is ambiguous, I assume this means the population where the individual was sampled in reality before being “artificially translocated”. But then it is stated that “location was specified as a categorical variable with four levels (…)” I really don’t understand how this explicative variable was coded. This needs to be better explained.

Experimental design

The authors propose an original procedure to validate the results of studies that estimate dispersal using genetic markers. In a nutshell, the validation consists in using the data to carry out “experimental” translocations of individuals among sampled populations and then carrying out assignment tests to estimate error rates. I think this is a useful way to evaluate the reliability of the results obtained by a given study and, therefore, it is a valuable contribution. However, the use of this procedure cannot prevent or correct errors in the sampling protocol used to obtain the dataset. This is something that should be discussed.

Validity of the findings

The findings of this study are consistent with those of previous studies but also provide some new insights (e.g. number of loci necessary for obtaining reliable estimates is larger than those suggested by previous studies).

Additional comments

Overall, this is a study that will help improve the quality of studies focused on dispersal. Besides the issues I highlighted above, there are two other main issues that I would like to authors to address.

a) Clearly, the proposed approach is only aimed at verifying that the dataset and method of choice can indeed provide reliable results. This is fine but ideally, one would like to have some idea about the sample sizes and number of loci needed to obtain reliable estimates before carrying out the field work needed to collect the samples. The proposed procedure cannot address this problem, only a preliminary “true” simulation study (i.e. based on synthetic samples) can help in this regard particularly in terms of sample sizes needed. I think the authors should discuss this issues in much more detail.

b) The literature search aspects of the study are poorly addressed in the current version of the manuscript. For example, it is not completely clear why the authors want to carry out an analysis to investigate the relationship between genetic differentiation and number of loci used. This only becomes apparent later on (c.f. lines 276-279). The authors should explicitly state the motivation for this analyses much earlier on (e.g. lines 174-176).

Other comments:

lines 267-269: How can you explain the fact that a locus with high PIC leads to erroneous results?

·

Basic reporting

This ms. is clearly written and carefully proofread (for which I'm grateful!); lit references and background are adequate and the authors clearly describe the limitations of their analysis, with some minor exceptions noted in my comments to the authors. Raw data are available and the results are relevant.

Fig. 4 and its associated analysis are perhaps a response to previous reviewers arguing that the manuscript is no longer relevant since other methods of identifying immigrants (not based on microsats) are available. I agree with the authors that microsat-based analyses are likely to continue, and so don't think this analysis and figure is necessary.

Experimental design

This manuscript addresses an important issue related to the use of microsatellite loci to identify interpopulation dispersers via genetic assignment tests. Such tests have long been known to "pick up" dispersers' descendants as well as the dispersers themselves (and of course there may also be "false negatives" and "false positives" of the sort expected in any statistical comparison). But few users have noted or attempted to evaluate the magnitude of this problem as it applies to their particular study system. This manuscript describes one approach to doing so, and it does a good job. I have two suggestions that I think could increase its value to readers:

1) As the authors note, their analysis applies only to their 3 populations, or perhaps to another set of populations with a similar level of differentiation. The conclusion that "at least 27 loci are needed" would not, for example, apply to a system with higher Fst. In this or future publications, I'd encourage the investigators to provide readers with more guidance about how to deal with the issue themselves.

In particular, could they provide the R code that an investigator dealing with a different set of microsats, or a different set of genotypes, would need to generate the equivalent of figure 2? Or, if they are suggesting the more "seat of the pants" approach outlined in fig. 5, can they suggest either how to choose the "test dispersers" in their simulation approach so as to maximize the possibility that they include the worst-case individual?

2) As the authors indicate, dispersers can be misidentified in two ways, false positives and false negatives. The paper would be more useful if the two types of misidentification were addressed explicitly. As I understand it, the response variable in the statistical model (and fig. 1) specifically addresses only the problem of correctly identifying dispersers as such (lines 149-152). Is it possible to generate a parallel analysis (and plot) dealing with the probability of correctly identifying a nondisperser? In other words, if I started with the microsat genotype of a an individual that was actually born in population A, how many loci would it take to be sure (at some level of probability) that GeneClass would not misclassify it as an immigrant to that population?

Validity of the findings

The analyses seem to me appropriate and clearly described, with some minor caveats noted in comments to the author. See comments above.

Additional comments

In addition to the major concerns listed in section 2, here are some secondary issues --

Line 67 fix font of “populations”

Line 70-71. Another extensive comparison of mark-recapture vs. assignment test vs. parentage analysis approaches to identifying dispersers is Waser, PM and JD Hadfield. 2011 (How much can parentage analyses tell us about precapture dispersal? Molecular Ecology.20:1277-1288).

A more obscure reference that used simulations to explore a similar question is Waser, PM, D Paetkau and C Strobeck. 2001. (Estimating interpopulation dispersal rates. Pp 484-497 in JL Gittleman, SM Funk, DW Macdonald and RK Wayne, eds. Carnivore Conservation. Cambridge University Press).

By the way, a broader set of simulations exploring the probability that dispersing animals (and their descendents) are detected by assignment tests as a function of number of loci, mating system, and other population parameters is still available in an unpublished manuscript (still available at http://bilbo.bio.purdue.edu/~pwaser/genetix/download.html , where you should also be able to download the simulation code). Though >20 years old, if this effort seems of use to the authors I’d welcome their use of it.

Lines 106-107: “These data were combined..”

Line 114: “better than Fst at handling…”

Line 124: Clarify: was ONE randomly-chose individual in each population involved in all of the simulations? If so, then are the quantitative conclusions essentially based on an N of 3? Wouldn’t it be better to repeat this procedure with multiple individuals?

assigned Line 126: “ being simulated dispersers..”

Lines 131-136: hard to follow, this needs to be broken into several shorter sentences.

Line 141: “an individual can incorrectly be identified as a disperser..

Line 143: The term “misassignment” is inherently confusing (a misassigned individual is one that is assigned to its birth population, rather than the population it’s captured or sampled in, so if you’re interested in dispersal, it’s in fact correctly assigned… I’d urge the authors to purge all uses of this word from the ms, and just to delete this sentence. Instead, tell us how this analysis deals with false positives and false negatives (and preferably discuss both of them)

Line 144: The authors should consider expanding this sentence so as to more clearly separate what I think are two slightly different ways of thinking about the problem: correctly identifying an individual as a first-generation immigrant (rather than a “resident”), vs. correctly identifying its population of origin. Both are discussed in this manuscript, but I think many readers will not immediately recognize that they are not exactly the same question.

Line 145: “In our assessment of the ability of GeneClass2.0 to detect simulated migrants, assignment tests were repeated..”
Line 150: Would the results be different if the response variable were “population of origin correctly identified (yes, no)”?

Line 189ff; Mention the range of PIC values?

Line 205: see comments above about clearer (and separate?) analyses of the effect of number of loci on false negatives and false positives.

Line 206: “from its collection”

Line 215: “were needed to assign SD-B.. (use “assign” rather than “identify”

Line 218: the term “misassigned” is confusing, I’d delete this sentence. Or perhaps say something like “misassigned as a resident rather than as an immigrant”? More generally: this individual is puzzling and interesting – would simulating dispersal by a broader range of individuals indicate how often this pattern occurs?

Line 226: delete ‘up”

Line 231: excess “precision” given the sample size : “on average 11.2 +/- 5.1 loci.

Line 256 How did the authors arrive at these numbers (“at least 15”)? Can they suggest a quantitative approach, given their fig. 2?

Line 320: I appreciate the sentiment, but this sort of took me aback. Why were any starlings killed: was the sample a “byproduct” of some sort of control program? If so, mention briefly in the methods.

Fig. 1: Fig. 4: “Piry et al. 2004” -- but I’m not convinced this graph is necessary.

Fig. 5: I’m puzzled by step 1, “identify # of genetic populations”. In the authors study, thy started with 3 samples, which (like most investigators) they label as “populations” without actually knowing whether they are in fact separate genetic populations in some idealized population genetics sense. In contrast, some assignment routines attempt to infer separate populations from the overall sample of genotypes – better to either delete this step from the figure, or explain it in the legend or text.

Overall -- a nice "seat of the pants" approach, perhaps not statistically sophisticated but far better than ignoring the problem!

Peter Waser

---

## Round 0.2 · accepted · Accept

· Academic Editor

Accept

I have now read your response to the reviewers and your revised article. You have addressed each of the concerns raised or clearly explained why you are not able to in both your rebuttal and the revised manuscript, so I am satisfied with your revisions. Therefore, I am happy to accept your manuscript and move it into production.